# Modular Approach for Odometry Localization Method for Vehicles with Increased Maneuverability

**DOI:** 10.3390/s21010079

**Published:** 2020-12-25

**Authors:** Chenlei Han, Michael Frey, Frank Gauterin

**Affiliations:** Institute of Vehicle System Technology, Karlsruhe Institute of Technology, 76131 Karlsruhe, Germany; michael.frey@kit.edu (M.F.); frank.gauterin@kit.edu (F.G.)

**Keywords:** localization, odometry, unscented Kalman filter (UKF), vehicle models, omnidirectional, modular approach

## Abstract

Localization and navigation not only serve to provide positioning and route guidance information for users, but also are important inputs for vehicle control. This paper investigates the possibility of using odometry to estimate the position and orientation of a vehicle with a wheel individual steering system in omnidirectional parking maneuvers. Vehicle models and sensors have been identified for this application. Several odometry versions are designed using a modular approach, which was developed in this paper to help users to design state estimators. Different odometry versions have been implemented and validated both in the simulation environment and in real driving tests. The evaluated results show that the versions using more models and using state variables in models provide both more accurate and more robust estimation.

## 1. Introduction

### 1.1. Background

The development towards more assistance and automation in vehicles has increased significantly in recent years. As a preliminary stage to fully automated driving, the combined use of a lane-keeping assistance system (LKAS) and a Stop&Go adaptive cruise control (ACC) is already possible with series components. Parking assistance and highway pilot are already permitted legally in Austria, and have been since January 2019 [1]. One of the most important prerequisites is a highly accurate and robust localization. With increasing levels of driving automation, the demand on localization and navigation also increases. Localization and navigation not only serve to provide positioning and route guidance information for users, but also are important inputs for vehicle control.

In addition to automation, the electrification of the automotive industry is continuously progressing. The added value of electric drives over combustion engines is clearly evident, as electric drives can be integrated directly into every wheel. Together with novel suspensions, which allow greater steering angles, maneuverability can be significantly improved [2,3]. In the project “OmniSteer”, which was funded by the German Federal Ministry of Education and Research (BMBF), a demonstrator vehicle with wheel individual steering and a ±90° wheel steering angle was developed [4]. The driving maneuvers in Figure 1 can be realized. Such novel vehicle configuration and omnidirectional maneuvers also demand new requirements for localization.

### 1.2. State of the Art

Localization methods in the automotive industry can generally be divided into two categories, according to Ref. [5]: global and relative localization (Figure 2).

Global navigation satellite systems (GNSS), such as GPS, are a recognized approach to obtaining absolute position [6]. Real-time kinematic (RTK-GPS) can even provide a position with centimeter-level precision [7]. However, it suffers from the bad signal condition in urban areas and the stability of GNSS will be degraded strongly due to poor sky view, building obstructions or multi-path reflections [8].

With landmark-based navigation, a LiDAR or other vision sensor detects the object, which is a landmark in the scenery, and tries to assign it to a previously saved map. Landmark-based navigation provides higher localization accuracy in a known environment. However, it is a relatively expensive solution (especially with LiDAR), and strongly depends on the environmental conditions [5].

Inertial navigation uses an inertial measurement unit (IMU), which records the three-dimensional values of vehicle acceleration and rotational speed [6]. The measurement is integrated in time to determine the position. Its advantage is the availability of an IMU, which is generally independent of external factors. Typical for inertial navigation is the accumulation error. Such an IMU, which can limit the position error drift to less than 1500 m in the first hour, costs around EUR 80,000.

Visual odometry (VO) calculates the relative transformation from one image to the next and finds the complete trajectory of the camera based on the images [9]. VO is free of errors that are dependent on terrain or vehicle parameters, but it also depends on the environment.

Wheel odometry (Abbr. odometry) is a method of estimating the position and orientation of a mobile system using the data from its propulsion system [10]. In the area of automotive engineering, measured variables from the chassis (wheel rotation and direction), steering system (steering wheel angle or wheel angle) and yaw rate sensor are commonly used. Since these sensors are already available in vehicles, there is no additional cost. Odometry is the most widely used localization method. It offers good short-term accuracy, is inexpensive, and allows high sampling rates [11,12,13].

The same as the inertial navigation, the basic idea of odometry is the integration of the incremental motion information over time, which inevitably leads to accumulation errors. Orientation errors lead to large lateral position errors, which increase proportionally with the distance travelled by the vehicle. Besides this, odometry is sensitive to unsystematic errors, such as slip, road unevenness, side wind, etc. Therefore, odometry is always fused with other localization methods to increase the accuracy and ensure the robustness of the fused localization results. Despite these limitations, most researchers [13,14,15,16,17,18] still agree that odometry is an important part of the localization system, and the navigation tasks will be simplified if the accuracy of the odometry can be improved [10].

In most papers, odometry is designed for vehicles with conventional front axle steering. The most popular vehicle models to calculate the vehicle position are the kinematic two-track model [17,19,20], the linear single-track model [21] and the kinematic yaw rate model [22,23]. However, kinematic models have their own limitations during maneuvers with large steering angles and high lateral acceleration. Besides, the conventional models cannot fulfill the requirement of vehicles with wheel individual steering.

This paper introduces an odometry localization method using unscented Kalman filter (UKF) for vehicles with wheel individual steering systems and an increased steering angle (till ±90°). Three vehicle models are designed to meet the new requirements of a wheel individual steering system. In addition, a novel modular approach is introduced and implemented to design a state estimator for odometry. Using this approach, different odometry versions with a combination of the three vehicle models are designed. Different odometry versions have been validated both in the simulation environment, with a validated vehicle model, and in real driving tests using a demonstration vehicle. Different kinds of omnidirectional parking maneuvers have been used for the validation. Three evaluation criteria are used to evaluate the results. 

The division of this work is organized as follows: In Section 2, the vehicle models and sensors are shown. The process to design the state estimator for odometry using the modular approach is introduced. The descriptions of the test vehicle, test maneuvers and validation environment follow in Section 3. The evaluation results for different odometry versions are shown and compared in Section 4. Section 5 concludes this paper.

## 2. Odometry Localization Method

### 2.1. Functionality of a State Estimator

Odometry can be classified as state estimation. The complete process of a state estimator is illustrated in Figure 3. Model, sensor and estimation method are the three important components of a state estimation. 

The estimation method is central to a state estimator. For linear systems, the most popular and effective state estimator is the Kalman filter (KF) [24]. For nonlinear systems, there are many variations based on the KF, two of the most common ones being the extended Kalman filter (EKF) and unscented Kalman filter (UKF). Both variations operate within the existing KF framework but use different approaches to handle nonlinearity. EKF uses an analytical linearization approach involving Jacobian matrices, while UKF uses a statistical approach called unscented transformation (UT) [25]. Due to the nonlinearity of the system in this paper and its simple usability, the UKF has been used as the estimator.

The estimation process consists of two steps: the prediction step and the correction step [26]. In the prediction step, the predicted state ${\hat{X}}_{k|k-1}$ and the estimate covariance $P_{k|k-1}$ are calculated from the state transition models, which describe the transitions between the temporally successive states $X_{k-1}$ and $X_{k}$. The indexing notation k|k−1 expresses the period from k−1 to k. The correction step follows the prediction step. Based on the predicted state ${\hat{X}}_{k|k-1}$, a virtual measurement ${\hat{Y}}_{k|k-1}$ can be calculated using the observation models, which describe the relationships between the state variable $X_{k}$ and the available measurement from the sensors $Y_{k}$. At the same time, the real measurements $Y_{k}$ are acquired by the sensors. The Kalman gain $K_{k}$ is calculated, by means of which a weighting of the virtual measurement and the real measurement takes place, so that a corrected state ${\hat{X}}_{k}$ is obtained. The sensor values used for the weighting are called observations. The sensor values can also be used as inputs in models directly.

### 2.2. Sensors

The sensors available on the demonstrator vehicle were used, namely the speed sensors and the wheel steering angle sensors of the four wheels, as well as the yaw rate sensor in the geometric center of the vehicle (see Table 1 and Figure 4).

### 2.3. Vehicle Models

The aim of the vehicle models is to describe the vehicle position and orientation by means of the measurable driving variables (wheel speed, yaw rate and steering angle). Since the odometry method should be real-time-capable, the vehicle models should not be complex. The meanings of the symbols are listed in Table A1 and Table A2 in Appendix B.

#### 2.3.1. Motion Model

The motion model describes the relationship between the vehicle position and the orientation between time step k−1 and k, which is shown in Figure 5a.

The motion model can then be described at the geometry center of the vehicle by
(1)$$\begin{array}{l} \left[\begin{array}{c} x_{k}\\ y_{k}\\ \theta_{k} \end{array}\right]=\left[\begin{array}{c} x_{k-1}\\ y_{k-1}\\ \theta_{k-1} \end{array}\right]+\left[\begin{array}{c} v_{k-1}\cdot \Delta t\cdot \cos\left(\beta_{k-1}+\theta_{k-1}+\omega_{k-1}\cdot \frac{\Delta t}{2}\right)\\ v_{k-1}\cdot \Delta t\cdot \sin\left(\beta_{k-1}+\theta_{k-1}+\omega_{k-1}\cdot \frac{\Delta t}{2}\right)\\ \omega_{k-1}\cdot \Delta t \end{array}\right]\\ =\left[\begin{array}{c} f_{x}\left(x_{k-1},\theta_{k-1},v_{k-1},\beta_{k-1},\omega_{k-1}\right)\\ f_{y}\left(y_{k-1},\theta_{k-1},v_{k-1},\beta_{k-1},\omega_{k-1}\right)\\ f_{\theta }\left(\;\theta_{k-1},\omega_{k-1}\right) \end{array}\right], \end{array}$$
where ${\left[\begin{array}{ccc} x & y & \theta \end{array}\right]}^{T}$ are the vehicle position and orientation in the global coordinate frame. Δt is the sample time. v, β and ω represent the vehicle velocity, the side slip angle and the yaw rate, respectively. They are unknown vehicle states and should be determined through a suitable sensor. For non-measurable states, mathematical correlations to the available sensors must be found, so that they can be estimated. The yaw rate can be measured by the yaw rate sensor. The vehicle velocity and the side slip angle are only measurable by expensive devices.

#### 2.3.2. Complementary Models

**Model-1:** Model for wheel velocity $v_{i}$

Normally, the vehicle velocity is calculated as the average of the wheel speeds from the non-driven and non-steered axle (usually rear axle). However, due to the wheel individual steering and the increased steering angle, this calculation is no longer valid. Therefore, each wheel must be considered individually. The kinematic relationship between each wheel velocity and the vehicle velocity is shown in Figure 5b, and can be described by
(2)$$\left[\begin{array}{c} v_{i,x}\\ v_{i,y}\\ 0 \end{array}\right]=\left[\begin{array}{c} v\cdot \cos\left(\beta \right)\\ v\cdot \sin\left(\beta \right)\\ 0 \end{array}\right]+\left[\begin{array}{c} 0\\ 0\\ \omega \end{array}\right]\times \left[\begin{array}{c} r_{i,x}\\ r_{i,y}\\ 0 \end{array}\right]=\left[\begin{array}{c} v\cdot \cos\left(\beta \right)-\omega \cdot r_{i,y}\\ v\cdot \sin\left(\beta \right)-\omega \cdot r_{i,x}\\ 0 \end{array}\right]$$
where $v_{i,x}$ and $v_{i,y}$ are the wheel velocities decomposed in the x and y direction in the vehicle-fixed coordinate frame. $r_{i,x}$ and $r_{i,y}$ represent the wheel contact points. i indicates the wheel position (i=fl,fr,rl,rr).

Equation (2) can be used to derive the relationship between each wheel’s velocity and the vehicle’s velocity:(3)$$\begin{array}{l} v_{i}=v_{i,x}\cdot \cos\left(\varepsilon_{i}\right)+v_{i,y}\cdot \sin\left(\varepsilon_{i}\right)\\ =v\cdot \cos\left(\varepsilon_{i}-\beta \right)+\omega \cdot \left(r_{x,i}\cdot \sin\left(\varepsilon_{i}\right)-r_{i,y}\cos\left(\varepsilon_{i}\right)\right)=f_{v_{i}}\left(v,\beta ,\omega ,\varepsilon_{i}\right) \end{array}$$
where $\varepsilon_{i}$ is the angle between the vehicle longitudinal axis and the vector of wheel velocity, which describes the actual moving direction of each wheel. This angle will be called wheel velocity angle in this paper, and it can be derived using the wheel steering angle $\delta_{i}$ and the tire slip angle $\alpha_{i}$ (tire slip angle is the angle between the tire main level and the vector of wheel velocity):
(4)$$\varepsilon_{i}=\delta_{i}+\alpha_{i}$$

**Model-2:** Model for side slip angle β

The side slip angle is the angle between the vehicle’s movement direction and the vehicle’s longitudinal axis [27]. Equation (2) can be expressed as:(5)$$\beta =\varepsilon_{i}-\arccos\left(\frac{v_{i}-\omega \left(r_{i,x}\cdot \sin\varepsilon_{i}-r_{i,y}\cdot \cos\varepsilon_{i}\right)}{v}\right)$$

This means that four side slip angles will be calculated from four wheel velocities $v_{i}$. Using the four β from each wheel, the following equation can be constructed:(6)$$\beta =\frac{1}{4}\cdot \sum_{i=1}^{4}\varepsilon_{i}-\frac{1}{4}\cdot \sum_{i=1}^{4}\arccos\left(\frac{v_{i}-\omega \left(r_{i,x}\cdot \sin\varepsilon_{i}-r_{i,y}\cdot \cos\varepsilon_{i}\right)}{v}\right)=f_{\beta }\left(\varepsilon_{i},v_{i},v,\omega \right)$$

**Model-3:** Model for wheel velocity angle $\varepsilon_{i}$

According to Equation (4), the wheel velocity angle $\varepsilon_{i}$ can be determined with knowledge of the tire slip angle $\alpha_{i}$. However, the indirect measurement of the tire slip angle is highly dependent on the accuracy of the tire model and the accelerometer. Therefore, it is not suitable for real-time applications. Based on Equation (2), the wheel velocity angle can also be described as
(7)$$\varepsilon_{i}=\arctan\left(\frac{v_{i,y}}{v_{i,x}}\right)=\arctan\left(\frac{v\cdot \sin\beta +r_{i,x}\cdot \omega }{v\cdot \cos\beta -r_{i,y}\cdot \omega }\right)\;=f_{\varepsilon_{i}}\left(v,\beta ,\omega \right)$$

### 2.4. Design of Odometry Localization Method Using Modular Approach

The following elements can be derived from Figure 3 to design a state estimator. They are state variables, sensor values (as observations or inputs), state transition models and observations models with their parameters. A modular approach to designing a state estimator is developed in this paper, which helps users to determine the essential elements mentioned above. The process of this approach is illustrated by the flow chart in Figure 6. The five elements to be determined are listed on the left side. This approach consists of three steps, including the preparation step 0. In every step, the actions for a certain element (e.g., for state variables) are described in the relevant row. Steps 1 and 2 should be repeated until the termination condition after step 2 is fulfilled. The black arrows indicate the process flow, the green arrows the signal flow.

**Step 0:** In this step, the resources for a state estimator will be prepared. The resources here include, on the one hand, the models, which may be used as the state transition model or the observation model, and on the other hand the available signals on board, which may be used as inputs of the models or as observations to take part in the correction step in Figure 3. 

During this step, users need to be familiar with the characteristics of the target system. The models prepared here should contain the relations between the variables to be estimated and the available signals. The available signals here can be sensor measurements or outputs of a controller.


**Round**
**1**


**Step 1:** In this step, the unknown variables will be defined as state variables. Usually, the unknown variables in Round 1 are the variables to be estimated.

**Step 2.1:** After defining the first state variables, state transition models will be selected among the prepared models in step 0 to describe the state variables. Models with outputs of the defined state variables can be used. These models are often differential equations. If there is no suitable one among the prepared models, users have to design a suitable model or use a random walk model instead. A random walk model means that the state variable will inherit its value from the last time step.

Subsequently, the unknown variables in the state transition models need to be known. If the unknown variables can be measured by sensors, then these sensor signals will be defined as inputs. It is also possible to use the defined state variables as the sources of the unknown variables in the models.

**Step 2.2:** The second part of step 2 is to select or design observation models. Observation models transform the state variables into a form of sensor measurements. The output of the observation models must be a variable, which can be measured by a sensor. This sensor measurement will be defined as an observation. In addition, observation models cannot be differential equations.

The sources of the unknown variables in observation models can be sensor signals or the defined state variables. It should be noted that at least one state variable must be used as the source of the unknown variable in an observation model. Otherwise, such an observation model does not correct the state variables and loses its function. If there is no suitable model, this step can be skipped.

After steps 2.1 and 2.2, if some unknown variables in the state transition models and observation models still have no sources, a second round is necessary.


**Round**
**2**


**Step 1:** Similar to step 1 in the first round, the unknown variables in the state transition models and the observation models, which have not had their sources identified in Round 1, will be defined as new state variables.

**Step 2:** It is same as step 2 in Round 1.

The whole process can stop, after all unknown variables have found their sources.

#### 2.4.1. Odometry Basic Version

Using this modular approach, several versions for odometry localization are designed. Figure 7 shows the basic version of odometry localization.

**Step 0:** The models and the available signals have already been introduced in Section 2.2 and Section 2.3.


**Round 1:**


**Step 1:** In the case of odometry localization, the variables to be estimated are the vehicle position and orientation. The first state variables are $X={\left[\begin{array}{ccc} x & y & \theta \end{array}\right]}^{T}$.

**Step 2.1:** After defining the first state variables, state transition models have to be designed to describe the state variables. Equation (1)’s motion model can be used to describe the vehicle position and orientation. The motion model is divided into two blocks, respectively, for the position and orientation. The blocks of the motion model and the state variables are connected by a double arrow line, because the motion model is a discrete differential equation and needs the value from the last time step to determine the actual time step. There are three unknown variables, v, β and ω, in Equation (1)’s motion model, in which the variable ω can use the yaw rate sensor signal. Therefore, the yaw rate sensor signal is defined as an input $\omega_{I}$ (the subscript I stands for the input). However, there are no accessible sensors to measure vehicle velocity v and side slip angle β directly. These two variables remain temporarily unknown. 

**Step 2.2:** The goal of this step is to design the observation models to describe the sensor signals through at least one state variable. However, this step can be skipped if no suitable model exists. Because GPS data are not planned to be used, there is no model to describe the sensor signals by the defined state variables ${\left[\begin{array}{ccc} x & y & \theta \end{array}\right]}^{T}$.

After steps 2.1 and 2.2, there are still two unknown variables in the motion model that have no source to provide them signals. A second round is necessary.


**Round 2:**


**Step 1:** Two remaining unknown variables v und β from the last round are defined as new state variables. The state variable vector becomes $X={\left[\begin{array}{ccccc} x & y & \theta & v & \beta \end{array}\right]}^{T}.$

**Step 2.1:** After defining the new state variable, the existing motion model can be completed by the newly defined v and β, which means the unknown variables in the motion model use the newly defined state variables v and β as sources. After that, state transition models for v and β will be designed. If the users do not want to have a complex system, it is recommended in this round to use the defined state variables and the available sensor signals as inputs for the state transition models, so there will not be too many new state variables. If it is difficult to find a suitable model, a random walk model can be used instead of a model with physical meaning. In Figure 7, the random walk model is represented by a diamond with the letter “R”. The connection between the random walk model and the state variable is also a double arrow line.

**Step 2.2:** According to model-1 (Equation (3)), wheel velocities can be expressed by the state variables v and β. Thus, the wheel speed sensor signals are defined as an observation $v_{O,i}$ (the subscript O stands for observation). Model-1 also requires the yaw rate and wheel velocity angle as inputs. Assuming that the tire side slip angle can be ignored here, then the wheel velocity angles are equal to the wheel steering angles. Just like the yaw rate signal that has been already defined as an input $\omega_{I}$ in Round 1, the wheel steering angle signals are also defined as inputs $\delta_{I,i}$.

This process can be finished after Round 2, because all formerly unknown variables are now known. Based on the blocks in Figure 7, the five essential elements that make up the state estimator can be formulated and written down in the right column.

Based on the basic version, some modifications have been made to optimize the state estimator in the following sections.

#### 2.4.2. Odometry Version 111

In the basic version, the yaw rate sensor was directly used in the motion model and model-1. However, the raw signal from the yaw rate sensor always contains noise and error. If the signal is used in models without processing, the error will be inherited by the output of the models. Although real measurements from sensors can be directly used for unknown variables in models, it is also possible to define the known variables (here the yaw rate ω) as state variables. A better result for the yaw rate can be reached through a suitable state transition model and correction by the yaw rate sensor. This also benefits the position and orientation estimation. Figure 8 shows the modified Version 111 based on the basic version. Dashed lines and dashed blocks denote the differences from the basic version.

With the new defined state variable ω, the motion model and model-1 will be updated. Steps 2.1 and 2.2 should be executed again. To simplify the system, the random walk model is used again in step 2.1 as the state transition model for ω. In step 2.2, the measurement from the yaw rate sensor can be directly expressed by the state variable ω. A passage model is inserted here and represented by a diamond with number “1”.

The right column of Figure 8 was modified according to the change. The differences from the Figure 7 are marked in blue.

Before introducing additional versions, the meaning of the version numbers is explained in Figure 9. There are three complementary models in total. The three numbers of the version number represent how the three complementary models have been used in the version, respectively. There are two states to describe the usage of model-1 and three states each for model-2 and model-3. Model-1 describes the relation between the vehicle states and each wheel’s speed, which is important for vehicles with a wheel individual steering system. Thus, model-1 must be used and the number for model-1 only contains the information of which input signal or state has been used. 

For example, Version 111 means that model-1 has used wheel steering angles, signal $\delta_{I,i}$, as input, while model-2 and model-3 are not used. The differences between $\delta_{I,i}$ and $\varepsilon_{i}$ will be explained in Version 212. 

The basic version is not numbered in this paper, because the yaw rate signal from an inertial measurement unit usually is not used directly without passing a filter.

#### 2.4.3. Odometry Version 212

Usually, assumptions are made to simplify model complexity. While using model-1 (Equation (2)) in the previous versions, we assumed that the wheel velocity angle equals the wheel steering angle, which does not meet reality, especially when the lateral acceleration is high. Thus, the wheel velocity angles $\varepsilon_{i}$ should also be treated as an unknown variable and defined as state variables, which is realized in Version 212 (see Figure 10). In this version, a random walk model is used as the state transition model at first. For the observation model, a passage model is used, although the wheel velocity angle does not equal the wheel steering angle, and the difference between the two variables can be modeled as measurement noise.

After defining the wheel velocity angles as state variables, model-1 can use either the state variables $\varepsilon_{i}$ or the inputs $\delta_{I,i}$. In Version 212, the new state variables $\varepsilon_{i}$ are used in model-1.

#### 2.4.4. Odometry Version 232

There are altogether four random walk models in Version 212. The more random walk models there are, the worse the stability of a state estimation, because the state variables cannot be constrained in a reasonable range by physical models. Thus, model-2 (Equation (6)) was used in Version 232 to replace the random walk model for β (see Figure 11). Because the second term of Equation (6) has less effect compared to the first, the second term is not implemented. This model can use the inputs $\delta_{I,i}$ or the state variables $\varepsilon_{i}$. The state variables $\varepsilon_{i}$ are used in this version.

#### 2.4.5. Odometry Version 213

Similar to the idea in Version 232, model-3 (Equation (7)) is used in Version 213 (see Figure 12) in order to keep the state variables within a reasonable range. Unlike Version 232, which limits β directly through model-2, model-3 uses ω, v and β to calculate $\varepsilon_{i}$. $\varepsilon_{i}$ will be corrected by the observations $\delta_{O,i}$, so that ω, v and β will also be corrected.

#### 2.4.6. Odometry Version 233

Version 233 (see Figure 13) is a combination of Versions 232 and 233. All of the three models are used in this version.

#### 2.4.7. Comparison of Odometry Versions

A total of 14 versions have been designed and are listed in Table 2. The different versions are categorized according to the model they used.

The covariance matrices of system noise and measurement noise for different versions are determined empirically. 

## 3. Validation

### 3.1. Test Vehicle

In order to test the performances of different versions, test drives were carried out with the demonstration vehicle shown in Figure 14a. The demonstration vehicle was developed in project “OmniSteer” and uses a novel chassis system. This novel chassis system [4] in Figure 14b,c applies a concept of steering wishbones, so that a 90° steering angle can be reached in both directions. 

### 3.2. Driving Maneuvers

Omnidirectional maneuvers requiring increased steering angles usually appear during parking, especially when the leeway is not sufficient. Odometry is also an efficient and suitable method to estimate the vehicle position and orientation in parking maneuvers [21]. For these reasons, different odometry versions in this paper have been evaluated through omnidirectional parking maneuvers. Figure 15 shows the eight driving maneuvers considered. The red vehicles or bicycle are the obstacles.

### 3.3. Validation Environments

The validation of the odometry consists of two steps. Firstly, the 14 odometry versions have been tested in a simulation environment. Figure 16a shows the validation process in the simulation environment. A validated multibody dynamic model [28], including the novel suspension and steering system, has been used to generate the reference values. Noises and errors determined from the real sensors have been added into the signals of the virtual sensors. The artificial sensor signals serve as inputs for the odometry algorithm. The vehicle position and orientation estimated by the odometry are to be compared with the reference from the vehicle model. Figure 16b shows one of the omnidirectional parking maneuvers.

The second step is validation by real driving tests. To evaluate and validate the estimated localization results, the exact determination of the reference position of the vehicle is crucial. Since there is no centimeter-level GNSS on the test vehicle, two laser pointers have been mounted on the longitudinal axis of the vehicle. Before and after a test drive, the position was marked on the ground by use of the two laser pointers. With the help of the marked points, the real position and orientation are calculated (detail in Appendix A).

### 3.4. Robustness Analysis

A localization method must not only be accurate, but also robust. The 14 odometry versions have been examined to see if they are robust against the failure of one or more sensors. The sensor failures were generated artificially. The corresponding sensor signals were set to zero. A total of 14 cases have been defined (see Table 3). Robustness analysis was first carried out without detection of the failed sensor and then with detection by switching off the corresponding observation models.

### 3.5. Evaluation Criteria (EC)

To evaluate the performances of the 14 odometry versions, the following errors are considered:**EC-1:** End position and orientation error in global coordinate system (with superscript “*E*”):
(8)$$e_{pos,end}^{E}=e_{pos,t}^{E}=\sqrt{{\left(x_{t}^{E}-{\hat{x}}_{t}^{E}\right)}^{2}+{\left(y_{t}^{E}-{\hat{y}}_{t}^{E}\right)}^{2}}$$
(9)$$e_{ang,end}^{E}=e_{ang,t}^{E}=\theta_{t}^{E}-{\hat{\theta }}_{t}^{E}$$**EC-2:** Maximum position and orientation error in global coordinate system while driving:
(10)$$e_{pos,max}^{E}=\max\left(e_{pos,n}^{E}\right)=\max\left(\sqrt{{\left(x_{n}^{E}-{\hat{x}}_{n}^{E}\right)}^{2}+{\left(y_{n}^{E}-{\hat{y}}_{n}^{E}\right)}^{2}}\right)$$
(11)$$e_{ang,max}^{E}=\max\left(\theta_{n}^{E}-{\hat{\theta }}_{n}^{E}\right)$$
**EC-3:** Average error of position change and orientation change:
(12)$${\bar{e}}_{dp}=\frac{\sum_{n=1}^{t}\left|dp_{n}-d{\hat{p}}_{n}\;\right|}{\sum_{n=1}^{t}dp_{n}}$$
(13)$${\bar{e}}_{d\theta }=\frac{\sum_{n=1}^{t}\left|d\theta_{n}-d{\hat{\theta }}_{n}\;\right|}{\sum_{n=1}^{t}d\theta_{n}}$$



The meaning of the symbols in Equations (8) to (13) can be found in Figure 17.

## 4. Results and Discussion

### 4.1. Simulation Results

Figure 18 shows the trajectories of different odometry versions in two omnidirectional parking maneuvers (DM1 and DM3 of Figure 15) in the simulation environment. Most of the proposed odometry versions (except versions with only model-1, namely 111, 112 and 212) are able to estimate the vehicle position and orientation in the omnidirectional parking maneuvers.

To compare the accuracy of different odometry versions, the simulation results have been evaluated according to EC-2 and EC-3. The evaluation results are shown as a boxplot in Figure 19. In addition to the boxplot, the data used for the boxplot have been plotted in grey. Different odometry versions have been grouped according to the models used (the motion model is not included here, because the motion model is the basic model and necessary for every version). It should be noted that the ordinates (y-axis) are all logarithmic.

Figure 19a shows that the odometry versions equipped with more models provide better position estimations. The improvement is significant after the odometry is equipped with two models. The improvement does not look so obvious after using all three models. This can be explained through the diagrams created by the modular approach.

According to the motion model in Equation (1), v, β and ω are used to estimate the vehicle position. For the versions with only model-1, e.g., Version 111 in Figure 8, these three variables are not constrained by the state transition models with physical meaning. Thus, they can only be corrected by the sensors through the observation models. Figure 20a shows the correction paths of ω,  v and β of Version 111. ω will be corrected by the yaw rate sensor and by the four wheel speed sensors through model-1, while v and β can only be corrected by the four wheel speed sensors through model-1. Although it is enough to determine two variables using model-1 (Equation (3)) four times, if the vehicle moves straight with the same wheel steering angles, the four equations will be the same. v and β cannot be determined uniquely. This leads to an unstable position estimation.

With the introduction of model-2, the state variable β can be limited in a reasonable range, and the position estimation becomes stable. Figure 20b illustrates the correction step of Version 232. Similarly, v and β in the versions with model-3 (e.g., Version 213 in Figure 12) can be corrected by the wheel speed sensors through model-1 and by the wheel steering angles through model-3. The correction paths are shown in Figure 20c. The versions with model-3 have a more accurate estimation of position than the versions with model-2 because the second term of Equation (6) in model-2 is ignored. The accuracy of this model is reduced. For this reason, the accuracy of the versions with three models has not been improved significantly.

Compared to the position estimation, Figure 19b shows that the versions with less models have a better orientation estimation. According to the motion model in Equation (1), vehicle orientation is only determined by the state variable ω. In versions with model-1 and with model-1 and 2, ω is corrected by the yaw rate sensor and the wheel speed sensors through model-1. Because model-2 does not affect the orientation estimation, they have the same quality in orientation estimation. With the introduction of model-3, the estimation accuracy decreases slightly. The reason may be that model-3 makes no positive contribution, or that the covariance matrices are not optimal. 

EC-3 in Figure 19c,d shows the same tendency as EC-2 in Figure 19a,b.

After the accuracy assessment, the robustness of different versions was examined. This analysis was carried out according to Section 3.4. The results are shown in Figure 21. It can be seen that failure detection is necessary if sensor failures occur. The most robust Versions (213 and 233) can provide an absolute position error less than 0.3 m and an average position change error less than 0.2 m/m. Their orientation errors are all less than 3 deg and 20 deg/rotation.

Focusing on the results with error detection, the more models that are included, the more robust the results will be. Even in orientation estimation, although the versions with model-1 and with model-1 and 2 are better than the versions with model-1 and 3 and model-1, 2 and 3 in most cases, there are outliers, which are less accurate.

Figure 22 describes the robustness in other aspects. The influences of failures on four versions are shown individually. These four versions use the state variables $\varepsilon_{i}$ among the versions using the same models. They also have better robustness in their group. With regard to the position estimation, the versions with model-3 (213 and 233) can provide good estimation under all failures, while Version 212 (only with model-1) cannot withstand even one failure. Version 232 (with model-1 and 2) can still provide good estimation if one of the wheel speed sensors or the yaw rate sensor fails. However, the failure of the wheel steering angle sensors lets Version 232 give no more reasonable position estimation, because the error in the wheel steering signals will influence model-1 and model-2 directly. 

Regarding orientation estimation, although the versions with model-1 (212 and 232) can suppress the vehicle orientation error by less than 1 deg and 5 deg/rotation in most cases, these two versions cannot withstand the yaw rate sensor failure. In contrast, the versions with model-3 (213 and 233) can provide stable orientation estimations even if the yaw rate sensor fails, because model-3 also helps to correct ω.

### 4.2. Real Driving Results

Figure 23 shows the trajectories of different odometry versions in two omnidirectional parking maneuvers in the real driving tests.

The figures for evaluation are structured as in Section 4.1. Instead of EC-2 and EC-3, EC-1 has been used because a reference system is not available. Start and end positions have been measured and used for evaluation. 

The results of position estimation from the real driving tests are basically consistent with the simulation results. As can be seen in Figure 24a, with the use of model-2 and model-3, the position estimation has improved significantly. However, the improvement of using model-3 is not reflected in the real driving tests. There are two possible reasons for this. Firstly, since EC-1 replaces EC-2 and EC-3 to evaluate the odometry, the error at the end position cannot reflect the estimation accuracy truly. Another reason may be the calibration error of the wheel steering angle, which affects the contribution of model-3 in calculating $\varepsilon_{i}$. 

A reduction in the orientation accuracy with model-3 cannot be seen in the real driving tests; see Figure 24b. All the versions have almost the same accuracy. Because the first-used yaw rate sensor has a poor resolution of 0.1°/s, the negative contribution of model-3 to orientation estimation cannot be noticed. 

For this reason, a yaw rate sensor with a resolution of 0.01°/s has been implemented. Figure 25 shows the results after the implementation of the new yaw rate sensor. We can see a reduction in orientation accuracy in the versions with model-3.

Figure 26 and Figure 27 show the results of robustness analysis in real driving tests, which match the conclusions made in the simulation section.

## 5. Conclusions

This paper investigates the possibility of using an odometry method to estimate the position and orientation of vehicles with increased maneuverability in omnidirectional parking maneuvers. Specific vehicle models were designed, and the sensors have been identified. A modular approach has been developed and implemented to help determine the essential elements for a state estimator. Different odometry versions were designed with this approach, and validated both in the simulation environment and in real driving tests. 

During the real driving tests, the position errors of Versions 213 and 233 are always under 20 cm, and the angular error is under 0.3°. Since there are no comparable results for omnidirectional parking maneuvers in the literature, we have compared the results with the results in Ref. [13] for normal parking maneuvers. Compared to the average position error of 47 cm and to the angular error of 1.9° in Ref. [13], the error level of odometry in this paper is acceptable for parking maneuvers.

In addition, the following conclusions were made regarding different odometry versions:The greater the number of models used to constrain state variables, the higher the estimation accuracy, theoretically. In reality, it also depends on the quality and effective range of a model. Sensor quality is also a factor;The use of the state variables $\varepsilon_{i}$ or the sensor signals $\delta_{i}$ as inputs in model-1 and model-2 plays no role in the estimation accuracy. If the state variables $\varepsilon_{i}$ have state transition model-3, the robustness of the odometry will be ensured if the sensors are defective;The versions with model-1 and 2, model-1 and 3 and model-1, 2 and 3 have almost the same accuracy. Versions 213 and 233 are the two most robust versions. 


The advantages of the modular approach can be summarized as follows:It simplified the process of designing a state estimator. The elements for a state estimator can be easily derived from the diagram of this method;The contribution of a certain model to accuracy and robustness can be predicted using this approach;New models or new sensors can be easily implemented, and possible effects can be analyzed;It provides a clear overview of all used models and sensors. It helps users to manage different estimators.


## 6. Future Works

As mentioned in Section 1.2, odometry is not suitable to be used alone. A part of future works is to investigate the contribution of the odometry in this paper during the fusion with GPS. Another work in progress is the sensitivity analysis for the parameter of the vehicle. For example, a larger wheel diameter may reduce the accuracy of odometry, because the wheel rotation speed will be reduced under the same vehicle speed, so that the wheel speed sensor provides a speed signal with low resolution. 

## Figures and Tables

**Figure 1 sensors-21-00079-f001:**
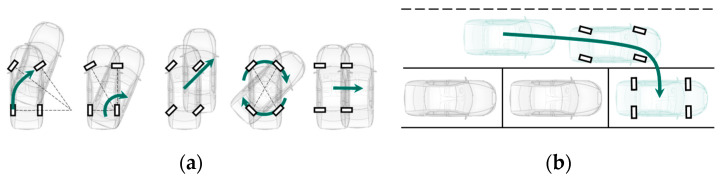
(**a**) Possible driving modes using novel suspensions; (**b**) possible continuously driven parking maneuver using novel suspensions.

**Figure 2 sensors-21-00079-f002:**
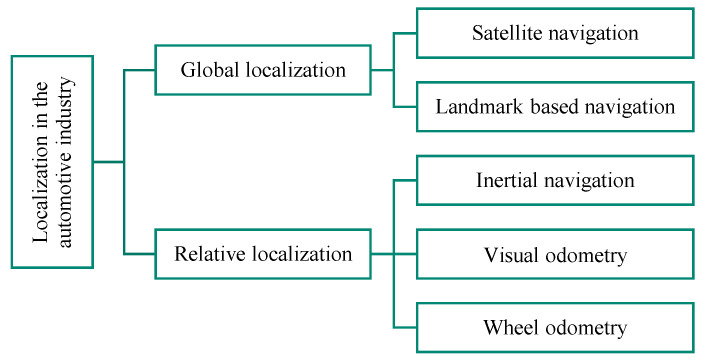
Classification of localization methods according to Ref. [5].

**Figure 3 sensors-21-00079-f003:**
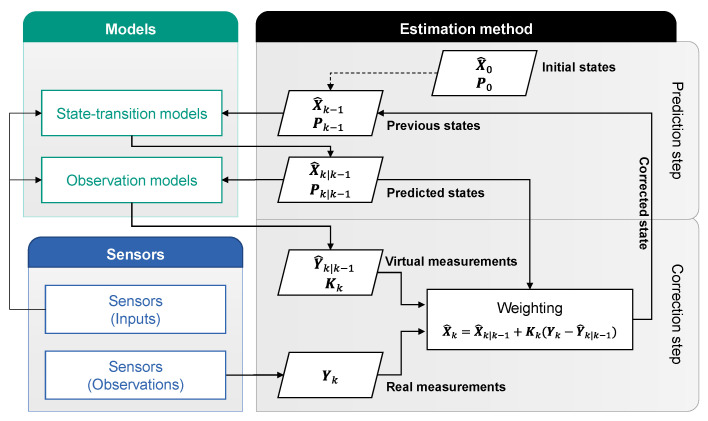
Functional sequence of a state estimator.

**Figure 4 sensors-21-00079-f004:**
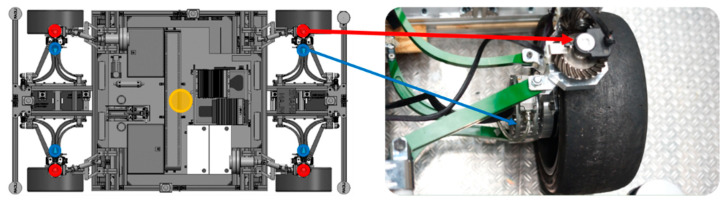
Position of the sensors (blue points: wheel steering angle sensors; red points: wheel speed sensors; yellow point: yaw rate sensor).

**Figure 5 sensors-21-00079-f005:**
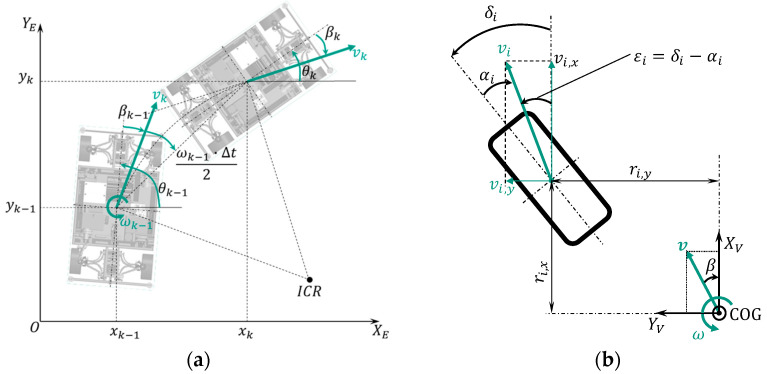
(**a**) Movement of a vehicle from time step k−1 to k; (**b**) kinematic relationship between wheel velocity and vehicle velocity.

**Figure 6 sensors-21-00079-f006:**
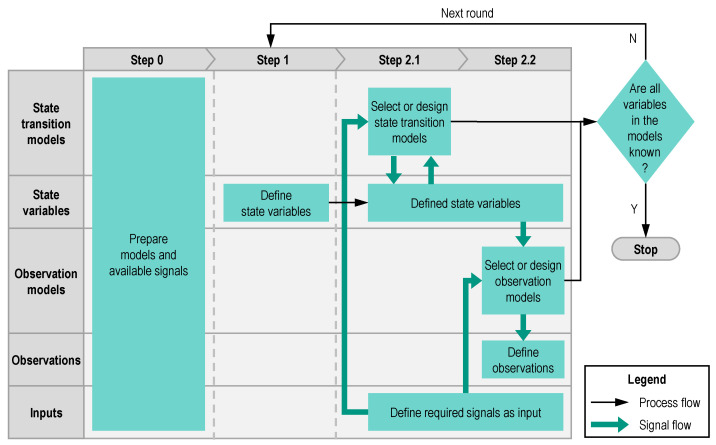
Flow chart of modular approach to designing state estimator.

**Figure 7 sensors-21-00079-f007:**
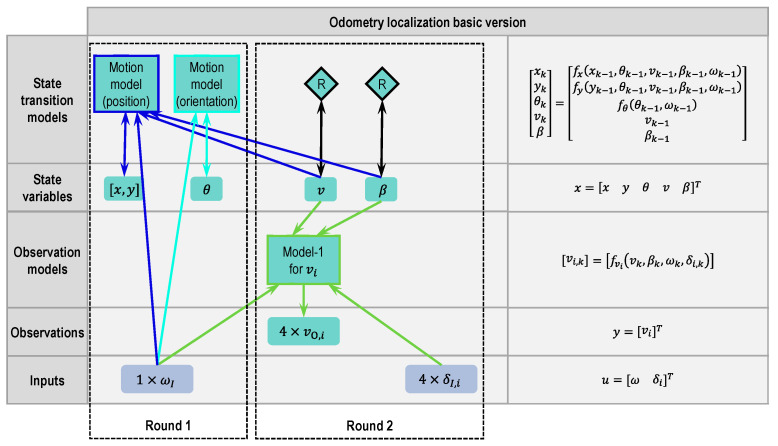
Modular approach to designing a state estimator for odometry localization (basic version).

**Figure 8 sensors-21-00079-f008:**
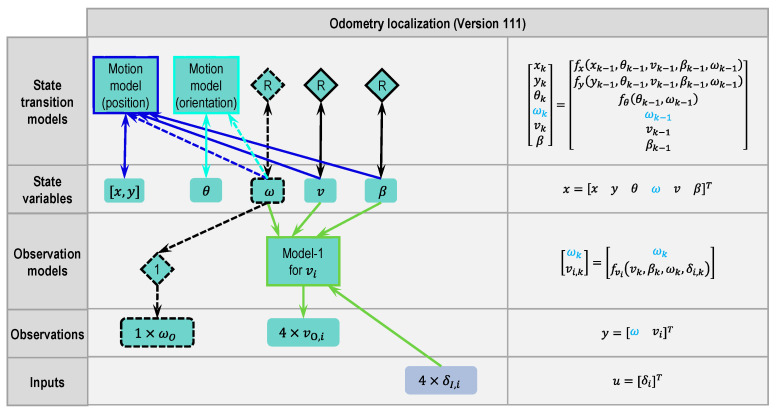
Modular approach to designing a state estimator for odometry localization (Version 111 based on basic version, changes marked with dashed lines or dashed blocks and blue letters, respectively).

**Figure 9 sensors-21-00079-f009:**
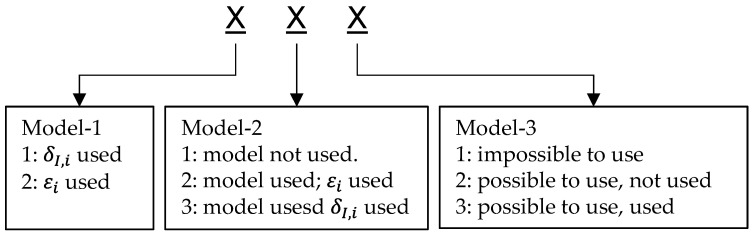
Explanation of version numbers.

**Figure 10 sensors-21-00079-f010:**
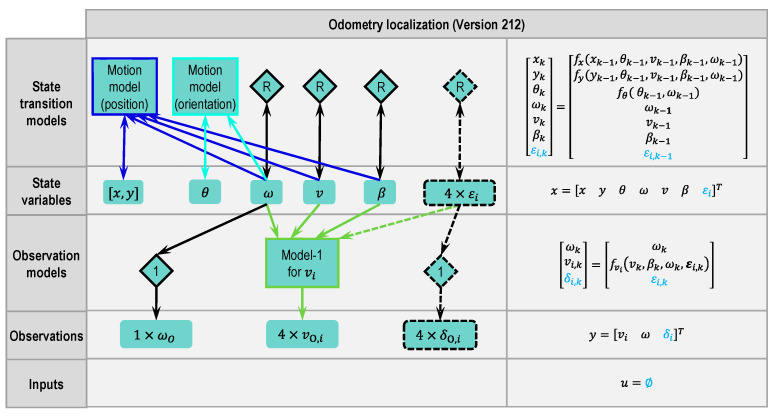
Modular approach to designing a state estimator for odometry localization (Version 212 based on Version 111, changes marked with dashed lines or dashed blocks and blue letters, respectively).

**Figure 11 sensors-21-00079-f011:**
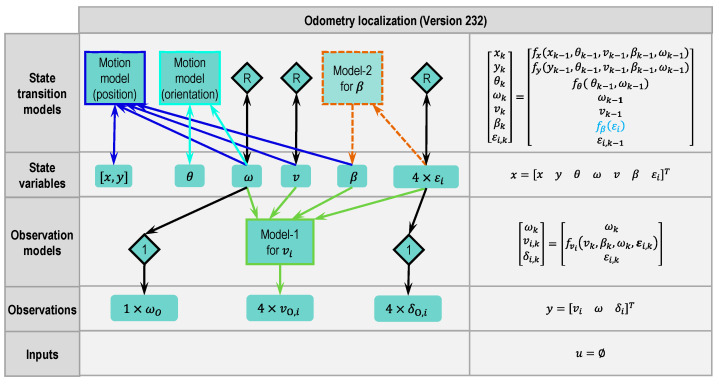
Modular approach to designing a state estimator for odometry localization (Version 232 based on Version 212, changes marked with dashed lines or dashed blocks and blue letters, respectively).

**Figure 12 sensors-21-00079-f012:**
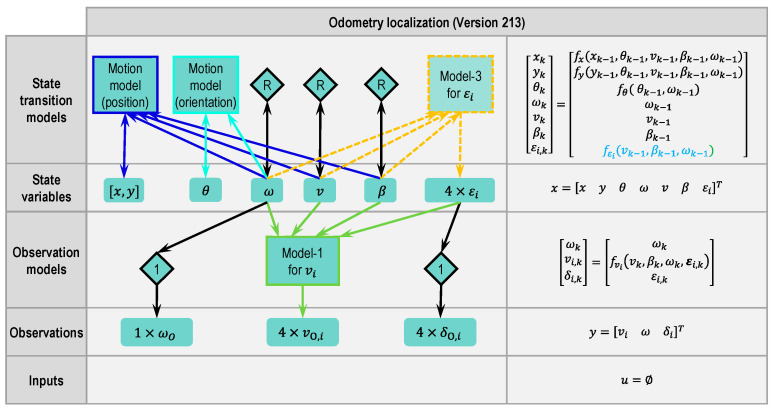
Modular approach to designing a state estimator for odometry localization (Version 213 based on Version 212, changes marked with dashed lines or dashed blocks and blue letters, respectively).

**Figure 13 sensors-21-00079-f013:**
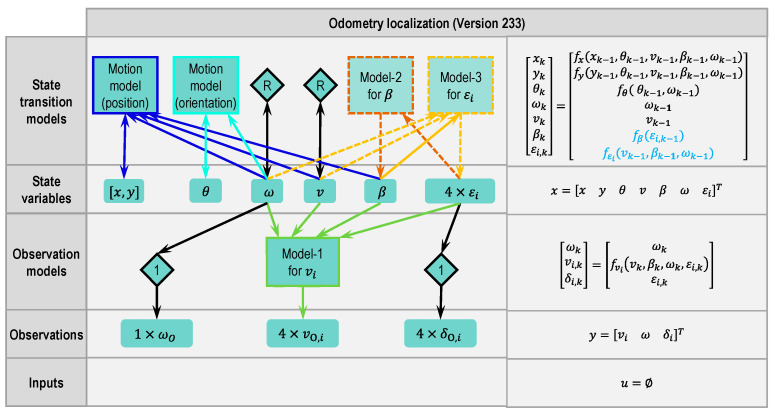
Modular approach to designing a state estimator for odometry localization (Version 233 based on Version 212, changes marked with dashed lines or dashed blocks and blue letters, respectively).

**Figure 14 sensors-21-00079-f014:**
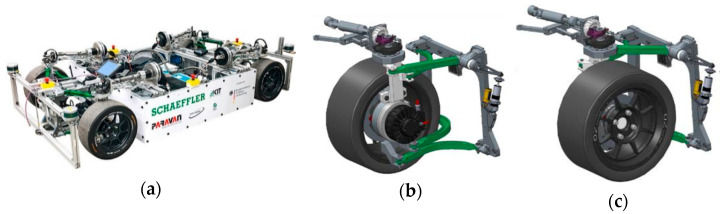
(**a**) Demonstration vehicle of project “OmniSteer”; (**b**) novel chassis system at increased steering angle 90°; (**c**) novel chassis system at increased steering angle −90°.

**Figure 15 sensors-21-00079-f015:**
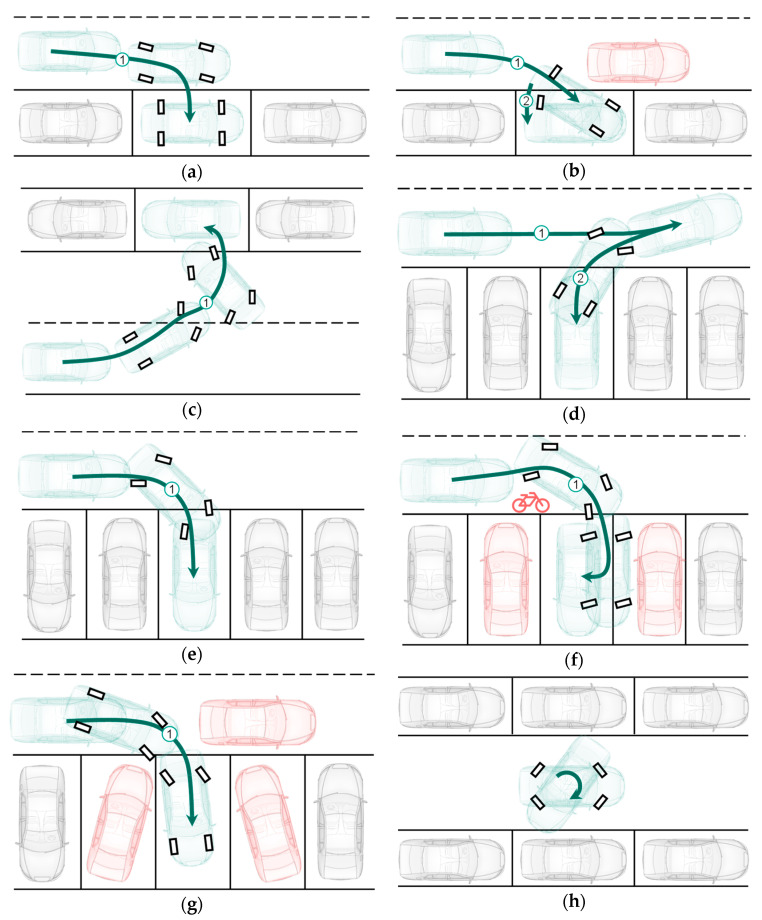
Driving maneuvers considered. (**a**) DM1: Vehicle parks in the parking lot without turning or stopping, 90° steering angle on each wheel can be realized; (**b**) DM2: Vehicle moves forward into the parking lot and the rear part of the vehicle is then pulled out. (**c**) DM3: Vehicle parks directly on the opposite side of the street without stopover; (**d**) DM4: Conventional reversing parking method; (**e**) DM5: Vehicle parks in a parking lot by steering the rear wheels in opposite directions to the front wheels; (**f**) DM6: Similar to DM5, but with correction at the end position; (**g**) DM7: Vehicle moves forward into the parking lot and the rear part of the vehicle is then pulled out, to avoid the obstacle; (**h**) DM8: Vehicle turns on its own vertical axis.

**Figure 16 sensors-21-00079-f016:**
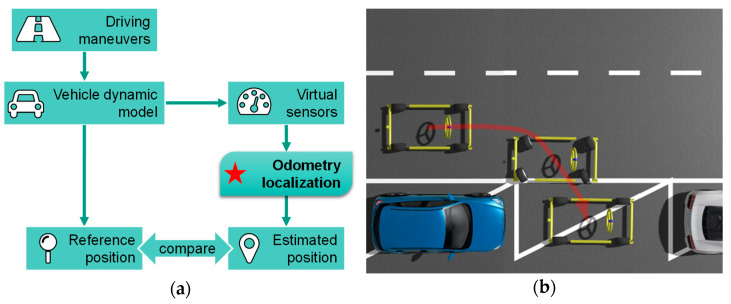
(**a**) Validation process in simulation environment; (**b**) virtualization of omnidirectional parking maneuver in simulation environment.

**Figure 17 sensors-21-00079-f017:**
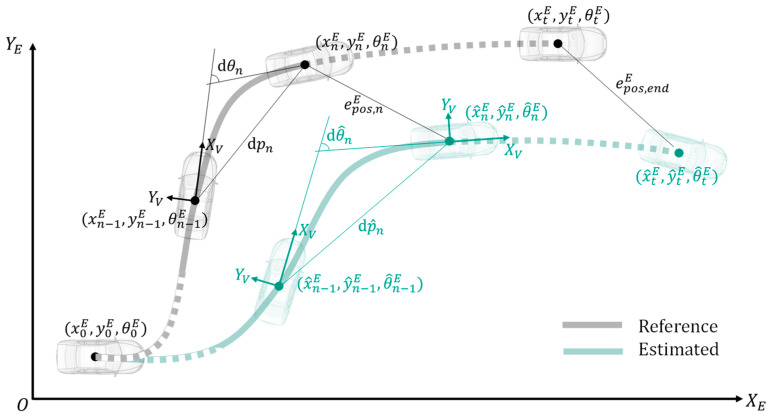
Schematic illustration of evaluation criteria.

**Figure 18 sensors-21-00079-f018:**
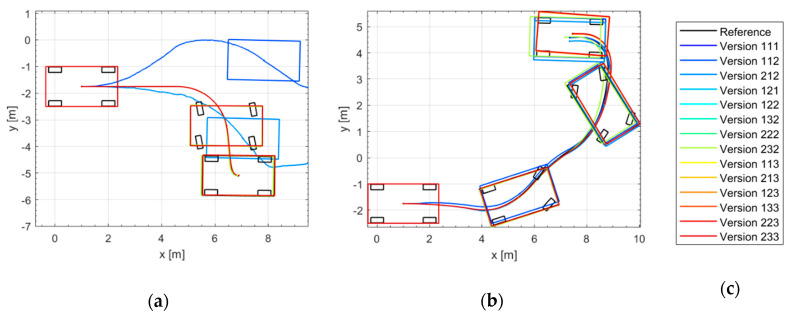
Simulation results of different odometry versions: (**a**) Trajectories in DM1; (**b**) Trajectories in DM3; (**c**) Legend.

**Figure 19 sensors-21-00079-f019:**
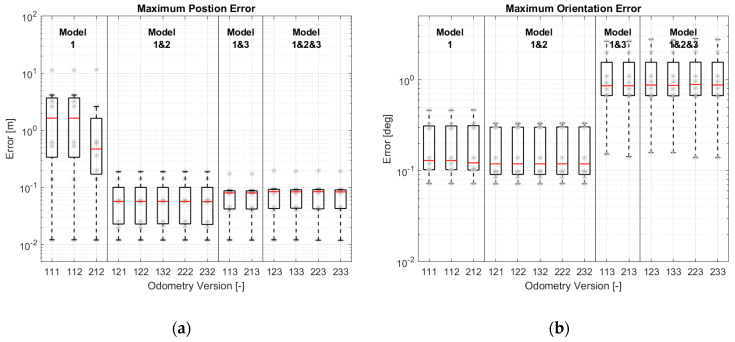
Simulations results of different odometry versions: (**a**) Maximum position errors; (**b**) Maximum orientation errors; (**c**) Average errors of position change; (**d**) Average errors of orientation change.

**Figure 20 sensors-21-00079-f020:**
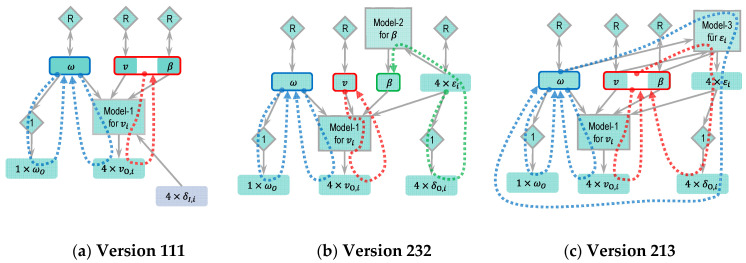
(**a**) Version 111: Correction paths for ω (blue), v and β (red). (**b**) Version 232: Correction paths for ω (blue), v (red) and β(green). (**c**) Version 213: Correction paths for ω (blue), v and β (red).

**Figure 21 sensors-21-00079-f021:**
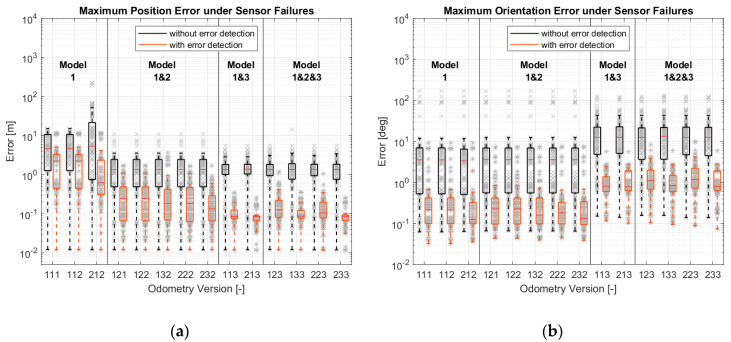
Simulation results of different odometry versions under sensor failures: (**a**) Maximum position errors; (**b**) Maximum orientation errors; (**c**) Maximum position errors; (**d**) Maximum orientation errors.

**Figure 22 sensors-21-00079-f022:**
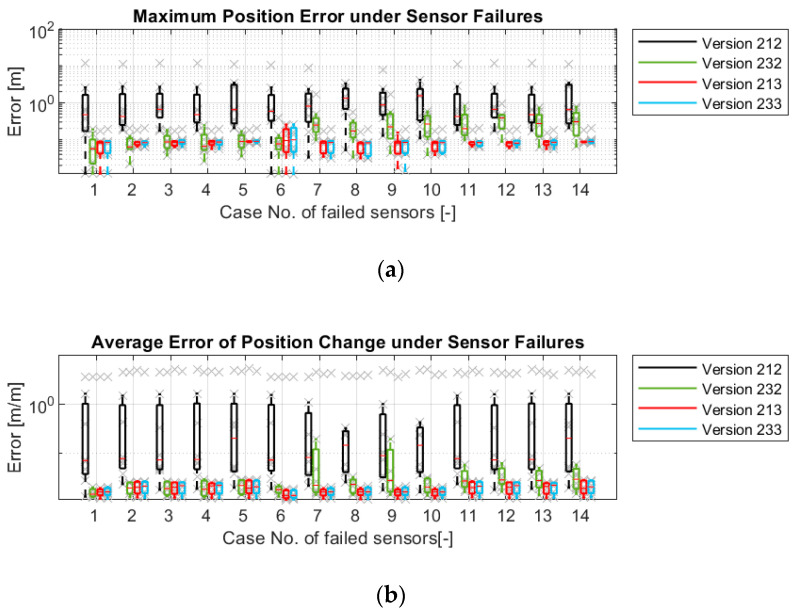
Simulation results of four odometry versions under sensor failures: (**a**) Maximum position errors; (**b**) Maximum orientation errors; (**c**) Average position errors; (**d**) Average orientation errors.

**Figure 23 sensors-21-00079-f023:**
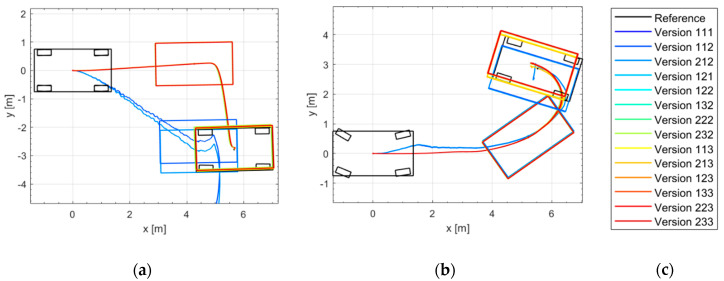
Real driving results of different odometry versions: (**a**) Trajectories in DM1; (**b**) Trajectories in DM3; (**c**) Legend.

**Figure 24 sensors-21-00079-f024:**
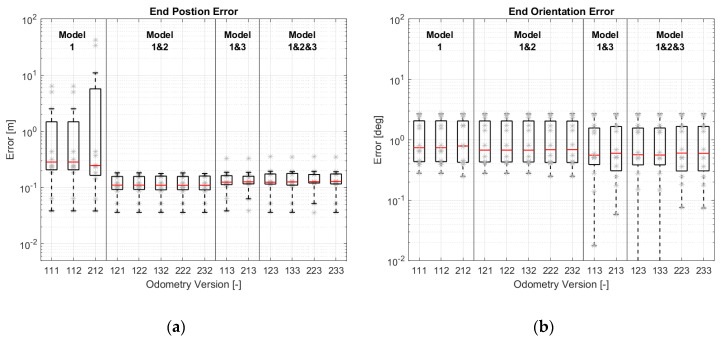
Real driving results of different odometry versions using a yaw rate sensor with a resolution of 0.1°/s: (**a**) End position errors; (**b**) End orientation errors.

**Figure 25 sensors-21-00079-f025:**
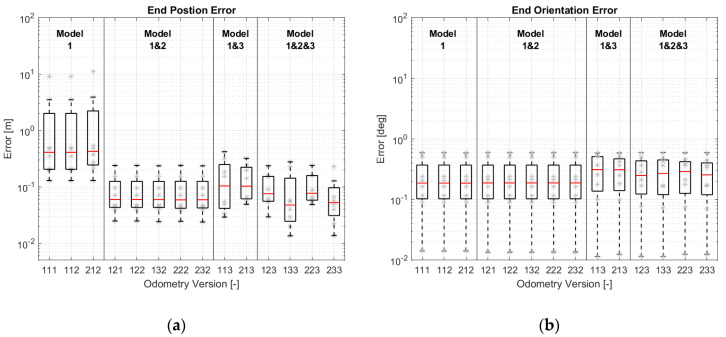
Real driving results of different odometry versions using a yaw rate sensor with a resolution of 0.01°/s: (**a**) End position errors; (**b**) End orientation errors.

**Figure 26 sensors-21-00079-f026:**
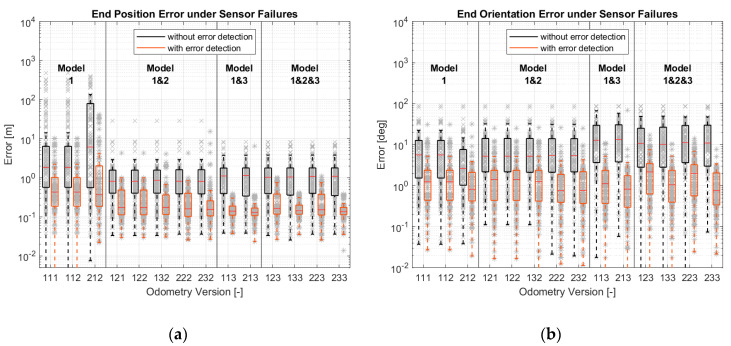
Real driving results of different odometry versions under sensor failures: (**a**) End position errors; (**b**) End orientation errors.

**Figure 27 sensors-21-00079-f027:**
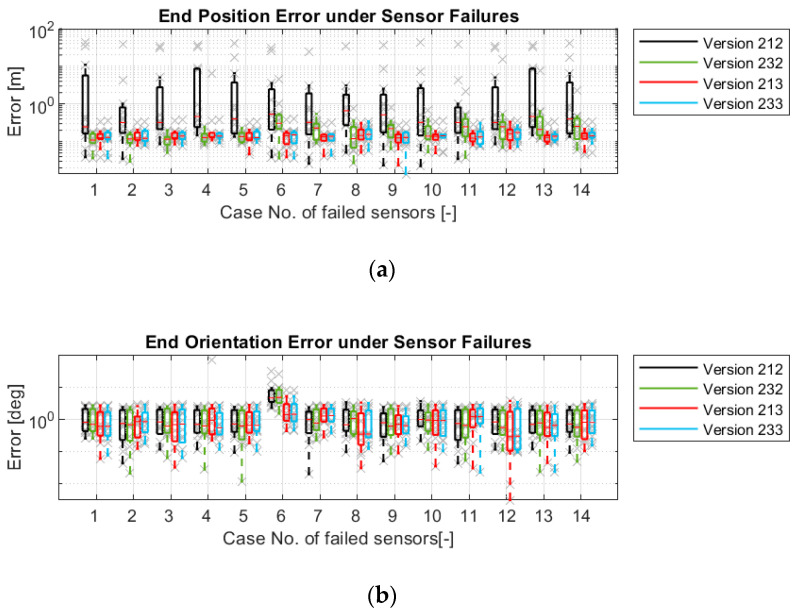
Real driving results of four odometry versions under sensor failures: (**a**) End position errors; (**b**) End orientation errors.

**Table 1 sensors-21-00079-t001:** List of the sensors on the demonstration vehicle.

Position	Sensor	Price/Unit
Wheel steering angle sensors	Bosch LWS 5.6.3	80 EUR
Wheel speed sensors	Integrated Speed Sensor of Traction Motor ^1^	-
Yaw rate sensor	UM7 IMU	150 EUR

^1^ Heinzmann PMS 080.

**Table 2 sensors-21-00079-t002:** Different odometry versions and the models they used.

**Version**	Model-1	Model-2		Model-3	
	$\varepsilon_{i}$ or $\delta_{I,i}$ Used?	Model-2 Used?	$\varepsilon_{i}$ or $\delta_{I,i}$ Used?	Model-3 Used?	
**111**	$\delta_{I,i}$	No	-	-	Model 1
**112**	$\delta_{I,i}$	No	-	No	
**212**	$\varepsilon_{i}$	No	-	No	
**121**	$\delta_{I,i}$	Yes	$\delta_{I,i}$	-	Model 1 and 2
**122**	$\delta_{I,i}$	Yes	$\delta_{I,i}$	No	
**132**	$\delta_{I,i}$	Yes	$\varepsilon_{i}$	No	
**222**	$\varepsilon_{i}$	Yes	$\delta_{I,i}$	No	
**232**	$\varepsilon_{i}$	Yes	$\varepsilon_{i}$	No	
**113**	$\delta_{I,i}$	No	-	Yes	Model 1 and 3
**213**	$\varepsilon_{i}$	No	-	Yes	
**123**	$\delta_{I,i}$	Yes	$\delta_{I,i}$	Yes	Model 1, 2 and 3
**133**	$\delta_{I,i}$	Yes	$\varepsilon_{i}$	Yes	
**223**	$\varepsilon_{i}$	Yes	$\delta_{I,i}$	Yes	
**233**	$\varepsilon_{i}$	Yes	$\varepsilon_{i}$	Yes	

**Table 3 sensors-21-00079-t003:** Cases to analyze the robustness of odometry.

Case No.	Failed Sensors
1	No sensor failed
2	Wheel speed sensor FL
3	Wheel speed sensor FR
4	Wheel speed sensor RL
5	Wheel speed sensor RR
6	Yaw rate sensor
7	Wheel steering sensor FL
8	Wheel steering sensor FR
9	Wheel steering sensor RL
10	Wheel steering sensor RR
11	Wheel speed sensor FL + Wheel steering sensor FL
12	Wheel speed sensor FR + Wheel steering sensor FR
13	Wheel speed sensor RL + Wheel steering sensor RL
14	Wheel speed sensor RR + Wheel steering sensor RR

## Data Availability

The data presented in this study are available on request from the corresponding author.

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
