# Peer review of "Modular Approach for Odometry Localization Method for Vehicles with Increased Maneuverability"

_sensors, 2020, doi:10.3390/s21010079_

Round 1

Reviewer 1 Report

The paper presents an interesting design and evaluation of a localization system of a vehicle with high maneuverability based on odometry with on-board sensors. That said, it is a bit confusing that the article is in the Remote Sensors section, since no external sensors are used. Well, it can be seen as a local alternative method to improve accuracy. Perhaps it would be interesting to include a comment on the possibilities of combining external and internal sensors, to take advantage of sensory fusion and improve robustness, or as a future task.

The role of sensors in the system is relevant. Three sensors have been used as shown in Table 1. Although it is a prototype, it would be interesting to mention the cost of the sensors to contextualize it with other methods indicated in the introduction.

The definition of epsilon_i is confusing considering Equation 4 and Figure 5.b. How is the tire slip angle alpha_i defined? According to the sense of alpha_i in Fig5.b it should be epsilon_i = delta_i + alpha_i. Please clarify.

In the simulation results, line 399, it would be convenient to modify the phrase "(except version 111, 112, 113)" by ("except versions with only model-1, namely 111, 112, 113)". Otherwise it is confusing in the description of the experiments.

Reviewer 2 Report

Very well prepared manuscript, nice graphics and text quality. After reading I found some remarks. Improving noted issues will improve readability and value of the paper.

  1. Lack of formulation of research, clear definition of errors for provided models and applied methods. This will help to improve conclusions, because then it is possible to evaluate quality and conditions of results.
  2. Explication of variables in equation I would like to see the separately described, maybe in the table. That will add clearness for reader.
  3. It would be nice to define influence of wheel diameter to odometry, at least in statement level.
  4. Lastly, acceptable linear and angular error level should be outlined in conclusions or discussion.
